# Preparation and Characterization of Disulfiram and Beta Cyclodextrin Inclusion Complexes for Potential Application in the Treatment of SARS-CoV-2 via Nebulization

**DOI:** 10.3390/molecules27175600

**Published:** 2022-08-31

**Authors:** Ana Maria Pereira, Ayse Kaya, Dan Alves, Niusha Ansari-Fard, Ibrahim Tolaymat, Basel Arafat, Mohammad Najlah

**Affiliations:** 1Pharmaceutical Research Group, School of Allied Health, Faculty of Health, Education, Medicine and Social Care, Anglia Ruskin University, Bishops Hall Lane, Chelmsford CM1 1SQ, UK; 2GMPriority Pharma Ltd., Priors Way, Coggeshall, Colchester CO6 1TW, UK

**Keywords:** disulfiram, solubility, β-cyclodextrin, SARS-CoV-19, nebulization

## Abstract

Disulfiram (DS), known as an anti-alcoholism drug, has shown a potent antiviral activity. Still, the potential clinical application of DS is limited by its low water solubility and rapid metabolism. Cyclodextrins (CDs) have been widely used to improve the solubility of drugs in water. In this study, five concentrations of hydroxypropyl β-cyclodextrin (HP) and sulfobutyl ether β-cyclodextrin (SBE) were used to form inclusion complexes of DS for enhanced solubility. Solutions were freeze-dried, and the interaction between DS and CD was characterized using differential scanning calorimetry (DSC), thermogravimetric analysis (TGA), and Fourier transform infrared spectroscopy (FTIR). In addition, the nebulization properties of the DS–CD solutions were studied. The aqueous solubility of DS increased significantly when loaded to either of both CDs. The phase solubility of both complexes was a linear function of the CD concentration (A_L_ type). Furthermore, physicochemical characterization studies showed a potent inclusion of the drug in the CD–DS complexes. Aerosolization studies demonstrated that these formulations are suitable for inhalation. Overall, the CD inclusion complexes have great potential for the enhancement of DS solubility. However, further studies are needed to assess the efficacy of DS–CD inclusion complexes against SARS-CoV-2 via nebulization.

## 1. Introduction

With over 500 million cases reported worldwide since the beginning of the pandemic and over 6 million deaths [1], the new coronavirus named severe acute syndrome coronavirus 2 (SARS-CoV-2) is responsible for a global viral outbreak of coronavirus disease in 2019 (COVID-19). Coronaviruses (CoVs), infecting many different animal species, can cause mild to severe respiratory tract infections in humans. In 2002 and 2012, respectively, severe acute respiratory coronavirus (SARS-CoV) and Middle East respiratory syndrome coronavirus (MERS-CoV), two highly contagious and pathogenic CoVs, emerged as pandemics causing many related deaths [2]. In late 2019, a novel CoV, showing a 79% identicality to the SARS-CoV genome and 50% to the MERS-CoV genome, emerged in the city of Wuhan, China [3].

The new data presented by World Health Organization suggested that vaccination significantly decreased COVID-19-related deaths and hospitalizations. However, the incidence of the cases is still significant; hence, the vaccination does not completely prevent COVID-19 from spreading again. In addition, conclusions on how long the immunity against the virus lasts are still unknown. Given the urgency of developing effective treatments, repurposing clinically approved drugs, off-patent and with a well-known mechanism of action, can be highly useful to obtain cheaper and safer drugs within a short period of time [4].

Disulfiram (DS) (Figure 1), a thiuram derivative, was approved by the Food and Drug Administration (FDA) in 1951 as an aldehyde dehydrogenase inhibitor to treat alcohol dependencies [5]. Given its limited toxicity and controllable side effects, DS is a valuable candidate for repurposing as a potential antiviral drug. Previous studies have shown that DS inhibits several enzymes, such as kinases, urease, and methyltransferase, indicating its potential antiviral activities with several mechanisms of action [6]. DS has been proposed as a potential HIV latency reversal agent [7] and an inhibitor of the non-structural protein 5A (NS5A) in the hepatitis C virus [8]. In addition, Min-Han et al. have demonstrated that DS successfully inhibited MERS and SARS coronavirus papain-like protease [2]. Furthermore, Jin et al. showed that DS was also able to act as a low molecular weight inhibitor of the M^pro^ viral proteases of SARS-CoV-2 [9]. Therefore, DS might act as a potential antiviral drug by reducing hyperinflammation in host cells and inhibiting viral proteases. However, the clinical application of DS is limited because of its poor water solubility [10], high instability in the acidic gastric environment, and rapid metabolism in the bloodstream [11]. Thus, there have been several attempts to tackle these limitations using nanotechnology [12,13,14].

Cyclodextrins (CDs) (Figure 2) are non-toxic cyclic oligosaccharides of glucose units produced by enzymatic degradation of starch [15]. These molecular structures are composed of 6, 7, or 8 α-D-glucopyranose units, known as parent cyclodextrins α-, β-, and γ-CDs, respectively. CDs are chemically and physically stable molecules characterized by their hydrophilic surface and lipophilic central cavity, which allow them to form complexes with a wide range of different molecules. Therefore, CDs have been widely used to improve solubility, dissolution rate, chemical stability, absorption, and bioavailability of poorly water-soluble drugs, as well as to reduce the side effects and toxicity of drugs [16]. Natural CDs have a relatively low solubility that limits their use in the pharmaceutical industry. Hence, they were subject to molecular modifications for enhanced solubility and increased central cavity size. For example, derivatives of β-CD present a properly sized central cavity for efficient drug loading and are easily produced at lower costs [15].

The most investigated or used types of β-CDs derivatives are the hydroxypropyl derivative (HP-β-CD), the sulfobutyl ether derivative (SBE-β-CD), and the randomly methylated derivative (RM-β-CD). However, HP-β-CD and SBE-β-CD are the most soluble; hence, they are the preferred CD derivatives used in pharmaceutical solutions [18]. Due to their unique capability to entrap guest molecules, CDs have been highly investigated to enhance the solubility of poorly soluble drugs via CD inclusion complexes. The structure of the CDs takes the form of a truncated cone or torus, with the primary hydroxyl oriented to the slimmer end and the secondary hydroxyl to the larger end. This makes the surface of the molecule relatively hydrophilic and the cavity hydrophobic [17]. In aqueous solutions, the water molecules inside the cavity are replaced by the drug forming CD-drug inclusion complex, yet without creating or breaking any covalent bonds [19,20]. The drug is stabilized inside the CD via different types of molecular forces, such as Van der Waals interactions or hydrophobic effects [21]. The formation of the inclusion complex of a poorly water-soluble drug allows the solubilization of this drug in aqueous solutions at higher (feasible) concentrations [22].

The use of CD inclusion complexes to enhance the solubility of DS is reported in the literature for the treatment of other diseases such as cancer [23,24]. However, only one type of β-cyclodextrin was used at high concentrations, and no phase solubility studies were performed. The aim of this study is to provide a critical assessment for using CD inclusion complexes to enhance the aqueous solubility of DS. Additionally, the potential use of CD–DS solutions to generate inhalable aerosols against SARS-CoV-2 was examined via nebulization.

## 2. Results and Discussion

### 2.1. Solubility Studies

The solubility of DS in a series of five different concentrations of each CD was examined. As shown in Figure 3a, the solubility of DS increased significantly when combined with the β-CDs, reaching more than 10 mg/mL at 20% *w*/*w*. For both CDs, the higher the concentration of CD, the higher the final concentration of DS (Figure 3).

The phase solubilities of DS with each CD (Figure 3b) show that the aqueous solubility of DS is a linear function of each CD concentration, suggesting the formation of inclusion complexes of A_L_ type. Hence, the total solubilized amount of the drug increases as a function of the CD concentration. Higuchi and Connors developed models classifying the complexes based on how the solubility of the drug behaves with the increasing concentrations of CD [25]. Accordingly, the most common type of cyclodextrin inclusion complexes is the 1:1 drug/cyclodextrin complex (D/CD) [26], meaning that one drug molecule (D) interacts with one cyclodextrin molecule (CD) to produce the D/DC complex (Equation (1)):(1)D+CD ↔D/CD

However, the Slope value, defined in the phase solubility diagrams, was less than a unit (Figure 3b) for both CDs. In detail, the molar ratios of the inclusion complexes were 1:2.5 and 1:3.4 DS:CD for SBE-CD and HP-β-CD, respectively. Wang et al. reported similar results for the solubility of DS with HP-β-CD, where a linear relationship (A_L_ type) was shown [27]. Yet, the results from this study were obtained over a wider range of concentrations.

To study the affinity of DS to each CD, the stability constants (K_1:1_) of both inclusion complexes were calculated using the following Equation (2) [22]:(2)$$K_{1:1}=\frac{Slope}{S_{0}\;\left(1-Slope\right)}$$
where S_0_ is the intrinsic solubility of DS.

K_1:1_ can be either determined using the intrinsic (equilibrium) solubility (S_0_) in water in the absence of CD or by considering the intercept (S_int_) from the phase-solubility diagram. Interestingly, for drugs with an aqueous solubility <0.1 mM, the S_0_ is usually larger than the S_int_, showing a negative intercept deviation for some complexes [26]. Accordingly, this phenomenon is seen in our results for DS SBE-β-CD inclusion complexes. Table 1 shows the results of the stability constants calculated from Equation (2), using S_0_ and S_int_.

It has been reported that the K_1:1_ value for a stable complex is within the range of 100–1000 M^−1^ [22]. Correspondingly, K_1:1_ calculated using S_0_ suggest that both CDs produce stable inclusion complexes with SD. Wang et al. reported a similar result using the intercept to calculate the stability constant of DS with HP-β-CD [27]. However, the discrepancies between S_0_ and S_int_ results show that this method might not be reliable for studying the affinity of poorly water-soluble drugs with different excipients [22]. A more accurate method was suggested using the complexation efficiency (CE) (Equation (3)) [22]:(3)$$CE=S_{0}\;K_{1:1}=\frac{\left[D/CD\right]}{\left[CD\right]}=\frac{Slope}{1-Slope}$$
where [D/CD] is the concentration of the inclusion complex and [CD] is the concentration of the free CD.

A CE of 0.64 and 0.42 were reported for SBE and HP-β-CD, respectively, (Table 1), suggesting that approximately three out of five molecules of SBE-β-CD and two out of five molecules of HP-β-CD form inclusion complexes with DS [22].

### 2.2. Physicochemical Characterization Studies of Freeze-Dried Formulations

#### 2.2.1. Differential Scanning Calorimetry (DSC) Analysis

The DSC thermograms for the free drug, raw CDs, and freeze-dried formulations are presented in Figure 4.

As shown in Figure 4a,b, DS powder has two sharp endothermic peaks at 74.2 °C and 208.9 °C, whilst no endothermic sharp peaks were detected in any of the pure CDs nor inclusion complexes, proving an efficient inclusion of the drug. Moreover, the peak detected at about 74.2 °C might correspond to the melting point indicating its crystalline state, whereas the peak detected at about 208.9 °C corresponds to its thermal degradation [24,28]. It was also observed in Figure 4 that the freeze-dried formulations (HP20 and SBE20) demonstrated similar profiles to that of raw CDs. Nonetheless, the absence of the sharp peak (at 74.2 °C) for the “guest” molecule (i.e., DS) in both thermograms of the inclusion complexes might be attributed to a strong molecular interaction with its “host” (the cyclodextrins) [29]. Therefore, the amorphous state of the drug was incorporated into the inclusion complex, making it more soluble in water. Similar results were reported by Qu et al. for the HP-β-CD [23].

It is also shown in Figure 4 that both pure CDs have a broad peak at about 80.3 °C and 87.9 °C for HP and SBE, respectively, caused by the evaporation of absorbed water. However, none of the CDs went through thermal degradation up to 250 °C. The slightly broad peak at about 230.1 °C shown by the HP20 thermogram indicates an interaction between DS and HP-β-CD [16]. SBE20 also reported a similar peak at 222.5 °C but less accentuated. These peaks might be due to partial degradation of the drug.

#### 2.2.2. Thermogravimetric Analysis (TGA)

In the TGA technique, the mass of sample is measured as a function of increasing temperature. The observed mass changes can be related to thermal events such as decomposition or de-solvation [30]. Results of the mass changes of pure DS, raw CDs (HP and SBE), and inclusion complexes of DS in both CDs (HP20 and SBE20) are presented in Figure 5.

The sample of the free drug had a significant decrease in mass within the range of 160 °C to 220 °C as a result of a fast decomposition rate [31]. Both pure CDs experienced different behaviors. As seen in Figure 5a, HP had a very slight (insignificant) change until 240 °C. Conversely, SBE experienced a slight mass loss until 110 °C due to the evaporation of absorbed H_2_O and then an insignificant change between 110 °C and 250 °C.

The first stages of mass change for both freeze-dried formulations occurred at about 80–110 °C for HP20 (Figure 5a) and 80–210 °C for SBE20 (Figure 5b) due to the evaporation of absorbed H_2_O. Thereafter, there was no significant decrease in weight during the second stage (110–200 °C for HP20 and 100–210 °C for SBE20); and the third stage was a result of the thermal degradations at different points for both formulations (at about 200 °C for HP20 and 210 °C for SBE20).

From Figure 5a,b, it can also be concluded that both formulations showed less resistance to degradation than pure CDs but stronger than that of the drug itself. This might indicate a formation of CD–DS inclusion complexes requiring more heat to break [29]. More explicitly, the sharp peak of degradation detected for DS was not present in any of the freeze-dried formulations. Thus, both CDs serve as a good protector of the drug from adverse external effects by forming efficient inclusion complexes that are physically stable [24].

#### 2.2.3. Fourier Transform Infrared Spectroscopy (FTIR)

FTIR spectra of pure DS, raw CDs, inclusion complexes, and physical mixture of DS with each raw CD are shown in Figure 6.

DS spectrum shows a C-H stretching vibration at 2974 cm^−1^; CH_2_ and CH_3_ deformations from 1348 cm^−1^ to 1456 cm^−1^; bands at 1495 cm^−1^ and 1271 cm^−1^ corresponding to N-C=S and C=S bond stretching, respectively; and C-C skeletal vibrations between 1149 cm^−1^ and 1193 cm^−1^ [28]. As shown in Figure 6, both inclusion complexes had a similar profile to that of the CD and DS physical mixture, but the peaks representing DS were not significantly detected in the HP20 and SBE20 spectra. Qu et al. and Tyukova et al. have also reported similar results for the HP-β-CD DS inclusion complexes [24]. This might be due to the embedment of the DS into the central cavity of the CD, indicating a successful preparation of both inclusion complexes and efficient protection of the drug against external factors [24]. Additionally, Suliman et al. reported that a similar “disappearance” of DDC-Cu vibrations bands in the inclusion complexes spectra was due to content lower than the limit of detection of the FTIR kit used [29].

Nonetheless, some of the most characteristic peaks of DS described above (C-C, C=S, CH_2_, CH_3_, and N-C=S) were still slightly detected in the freeze-dried formulations (1200 cm^−1^, 1270 cm^−1^, 1420 cm^−1^, and 1491 cm^−1^ for HP20; 1270 cm^−1^, 1351 cm^−1^, 1417 cm^−1^, and 1514 cm^−1^ for SBE20). This indicates that the chemical structure of the drug did not change during the preparation of the complexes [31]. Furthermore, the fact that Suliman et al. did not significantly report the DDC-Cu characteristic peaks with the same CDs might be due to the lower solubility achieved for DDC-Cu (4 mg/mL) [29].

It is worth mentioning that fused peaks of the drug within the exact same region were more detected for both physical mixtures, indicating that some of the DS did not enter the inner cavity of the CDs as it happens during the formation of the inclusion complexes with the freeze-drying technique, but contacted its outer surface [24]. However, further analytical studies such as XRD and NMR characterizations may be required to provide clear evidence of inclusion.

### 2.3. Determination of Total Aerosol Output, Droplet Size, and Fine Particle Fraction (FPF)

Appling energetic force, nebulizers convert liquid formulation into aerosols of a droplet size that can reach the lower respiratory tract. There are three types of medical nebulizers: air jet nebulizers, mesh nebulizers, and ultrasonic nebulizers. In this study, a Pari LC Sprint air-jet nebulizer was used. The mechanism of air-jet nebulizers is based on utilizing air pressure to force the liquid through the nozzle producing the aerosol droplets [32].

As shown in Figure 7, aerosol outputs were relatively high for all formulations, with a noticeable but not significant increase for the loaded CDs (*p* > 0.05), except for HP 10% (*p* < 0.05). Najlah et al. reported that air jet nebulizers usually give lower outputs compared to that of other nebulizers [33]. Thus, further studies using other types of nebulizers might be needed to confirm such differences. Nonetheless, this is outside the scope of this study.

To understand the aerosolization properties of CD–DS, solutions of freeze-dried formulations were studied using laser diffraction. Droplets smaller than 5–6 µm are considered “respirable”, but this is not the only factor that may influence this notion [34]. The different breathing patterns of individuals and patients can also influence the efficiency of nebulization. Therefore, additional studies are important, such as the use of breathing simulators mimicking different breathing patterns.

Droplets of a size smaller than 2.15 µm are deposited in the alveolar region; droplets larger than 11.66 µm may only reach the oropharyngeal region (i.e., unsuitable for pulmonary delivery) [33]. As shown in Table 2 (VMD and % ≤ 5.4), results from this study suggest that all formulations are suitable for inhalation. However, only the loaded CDs are more likely to stay within the peripheral regions of the lung, as the empty CDs are more likely to be exhaled, being deposited in the alveolar region of the lungs due to their droplet size being <2.15 µm (*p* < 0.05). Moreover, droplet sizes were significantly larger for HP 15% than for the other loaded CDs (*p* < 0.05).

Table 2 also shows size distribution results (span). Nebulization efficiency is also dependent on aerosols’ lower polydispersity, as they promote delivery to the lower respiratory tract [33]. All loaded CDs had smaller span values compared to that of empty CDs (*p* < 0.05). It is also important to mention that when comparing each HP-CD concentration, the HP 10% size distribution was significantly lower (*p* < 0.05).

The fine particle fractions (Figure 8) were relatively high for all formulations and within the same range, which might indicate that the liquid breakdown is not dependent on the constitution of the formulations [35]. These results are consistent with the findings of VMD and span.

## 3. Materials and Methods

### 3.1. Materials

Disulfiram (molecular weight = 296.51 g/mole) was purchased from Acros Organics, Loughborough, UK. Hydroxypropyl β-cyclodextrin (molecular weight = 1555 g/mole) and sulfobutyl ether β-cyclodextrin (molecular weight = 2242.05 g/mole) were both obtained from Glentham Life Sciences, Corsham, Wiltshire, UK. HPLC-grade water and ethanol (absolute and 70%) were purchased from Fisher Scientific, Loughborough, UK.

### 3.2. Methods

#### 3.2.1. Solubility Studies

The effects of HP-β-CD and SBE-β-CD on the solubility of DS were assessed by phase-solubility analysis, as were the interactions of these excipients on DS solubility. For this, solubility measurements and the determination of the final concentrations were carried out by adding an excess amount of disulfiram in five different concentrations of HP and SBE (1%, 5%, 10%, 15%, and 20% *w*/*w*) using HPLC-grade water as a solvent. After 1 h of sonication at room temperature, the samples were agitated at 150 rpm for 3 days at room temperature using Stuart reciprocating shaker SSL2 (Cole-Parmer, Neots, UK), and all formulations were then centrifuged (Thermo Scientific Heraeus PIC017, Fisher Scientific, Loughborough, UK) for 10 min at 13,000 rpm and the supernatants transferred and filtered. The final solutions were centrifuged again under the same conditions, further diluted using HPLC-grade water, and finally analyzed using a UV spectrophotometer (Jenway 6305 Spectrophotometer, Staffordshire, UK) at 262 nm, using solutions of the respective empty CD as blanks. The intrinsic (equilibrium) solubility of DS was performed as described above but in HPLC-grade water only.

The Beer–Lambert law (Equation (4)) establishes a relationship between the absorption of a sample (*A*) and its concentration (c), using the molar absorptivity (ε) and the optical path (l). This equation is linear until absorption values up to a unity, which means that in the linear range, it is possible to calculate the concentration of the drug through its absorption [30].
(4)A=ε l c

Results were assessed as a mean of three replicates, and the final concentrations of the drug were finally measured based on a linear equation that was deducted from a standard calibration curve of DS in ethanol. All results were validated using HPLC analysis according to our method described previously [12]. Final data were treated and presented in Excel.

The stability constants for the complexation of DS with the CDs were calculated using the following Equation (5) [22]:(5)$$K_{1:1}=\frac{\mathrm{Slope}}{S_{0}\;\left(1-\mathrm{Slope}\right)}$$
where $S_{0}$ is the equilibrium solubility of DS in the absence of CD.

#### 3.2.2. Freeze-Drying/Lyophilization

With the goal of getting a porous, amorphous powder with a high degree of interaction between the DS and each CD, the freeze-drying technique was carried out. Thus, in order to prepare a total amount of 4 g of solid freeze-dried formulations for each 20% CD solution, an excess amount of DS was added. Samples were then sonicated for 2 h, agitated for 24 h at 150 rpm, and filtered as explained above. Ultimately, formulations were left for 2h at −20 °C, 24 h at −80 °C, and freeze-dried for 3 days under negative pressure of 0.06 ± 0.01 using a Lyotrap LTE freeze dryer (LTE Scientific Ltd., Oldham, UK).

#### 3.2.3. Differential Scanning Calorimetry (DSC) Analysis

Thermal characteristics of the solutions were determined by DSC to analyze the inclusion of complex formation and release of the drug during storage. For this, 3–5 milligrams of the free drug, HP, SBE, and each freeze-dried solution were weighted and crimp-sealed in aluminum-led pierced pans. Analysis was performed using a differential Scanning Calorimeter (214 Polyma, Netzsch, Selb, Germany), over a temperature range of 0–400 °C and at a heating rate of 10 °C/min, and under nitrogen gas flow (20 mL/min). An inert gas flow is important in order to generate reproducible and accurate results. Nitrogen is the most common purge gas used in this technique since it is easily available, inexpensive, and has a low thermal conductivity which does not affect the DSC experiments [36]. The resulting thermograms were analyzed using Proteus Analysis software (version 8) and presented in Excel.

#### 3.2.4. Thermogravimetric Analysis (TGA)

TGA was used to evaluate the moisture content in the freeze-dried samples by comparing them with the thermograms of the free drug and CDs. Using a TG 209 F3 Tarsus model (Netzsch, Selb, Germany), 3–5 milligrams of DS, HP, SBE, and each freeze-dried formulation were weighted and loaded into an open fused silica crucible suspended from a microbalance and heated from 25 to 300 °C. Results were analyzed using Proteus Analysis software (version 8) and presented in Excel.

#### 3.2.5. Fourier Transform Infrared Spectroscopy (FTIR)

FTIR experiments were performed using Spectrum Nicolet iSF FTIR Spectrophotometer (Thermo Scientific, Loughborough, UK), supported with iDF ATR, where a small amount of DS, HP, SBE, the freeze-dried formulations as well as physical mixtures of DS in the same ratio as the solubility concentration with each respective CD were loaded directly in the spectrophotometer without any treatment using a wavenumber range of 4000 to 650 cm^−1^. The resulting data were obtained using OMNIC 9 software and presented in Excel.

#### 3.2.6. Determination of Total Aerosol Output, Droplet Size, and Fine Particle Fraction (FPF)

Nebulization analysis was performed using Malvern’s Spraytec laser diffraction size analyzer (Malvern Instruments Ltd., Malvern, UK) and Pari LC Sprint air-jet nebulizer. For the purpose of quantifying the total aerosol output, the nebulizer was weighted before and after nebulization, and the output percentage was calculated as shown in Equation (6):(6)$$Output\;\left(\%\right)=\left(\frac{weight\;of\;liquid\;nebulized}{weight\;of\;liquid\;before\;nebulization}\right)\times 100$$

The mouthpiece of the air-jet nebulizer was positioned at 3 cm and perpendicularly to the laser beam, and 3 mL of each empty and loaded CD solution was placed into the nebulizer reservoir. The total amount of liquid nebulized was quantified by subtracting the weight after nebulization from the weight before nebulization. Each formulation was studied 3 times, and the nebulizer was carefully washed and dried before each experiment. Nebulization was carried out until the aerosol generation ceased completely.

For the determination of aerosol droplets’ size and size distribution, respectively, it was applied via a vacuum line with 30 L/min flow rate to move the aerosol across the beam, and the volume median diameters (VMD; 50% undersize) and Span (Equation (7)) were recorded.
(7)$$Span=\frac{90\%\;udersize-10\%\;udersize}{VMD}$$

Furthermore, droplet sizes ≤ 5.4 µm were also deducted to represent deposition in the peripheral regions of the lung [33]. These were multiplied by the output percentage to determine the fine particle fraction (FPF).

#### 3.2.7. Statistical Analysis

Statistical analysis was performed using Excel, and all data were expressed as mean ± standard deviation. Statistical significance was analyzed by the Student’s t-test or one-way ANOVA as appropriate. In all tests performed, a level of statistical significance of 95% (*p* < 0.05) was considered.

## 4. Conclusions

COVID-19, the disease that has been taking over everyone’s life and seems to be a never-ending story, has been a research target for many scientists all over the world. Breakthroughs have been made to tackle SARS-CoV-2, including DS influence on its mechanisms of replication. However, disulfiram’s poor solubility has been its main limitation, preventing its use in many clinical applications. Thus, a novel complexation method of DS into two different types of β-CDs was investigated with the purpose of improving the aqueous solubility of the drug. The solubility studies performed with this aim, using different techniques, elucidated a significant improvement of DS’s solubility with both β-CDs. Physicochemical characterization studies of the freeze-dried formulations further confirmed their effectiveness, showing a potent inclusion of the drug in the CD–DS complexes. The high aerosol output and FPF and the small sizes and polydispersity suggest that these inclusion complexes are promising carriers for pulmonary delivery by nebulization. The next step for this research would be to perform a comparison study against other types of nebulizers. In addition, an in vitro biological activity evaluation of the inclusion complexes against SARS-CoV-2 is needed.

## Figures and Tables

**Figure 1 molecules-27-05600-f001:**
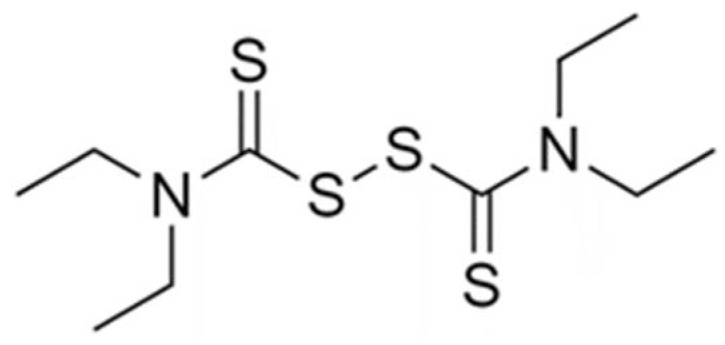
Chemical structure of disulfiram (DS).

**Figure 2 molecules-27-05600-f002:**
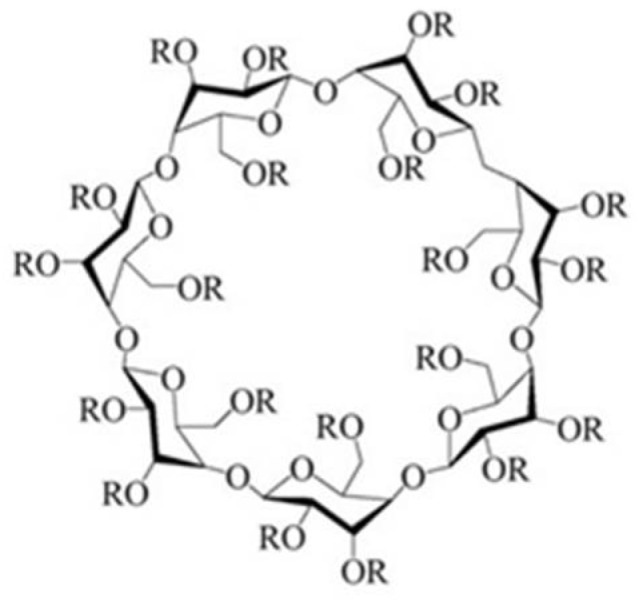
Chemical structure of hydroxypropyl β-cyclodextrin and sulfobutyl ether β-cyclodextrin, adapted from [17]. **HP-****β****-CD:** R = -CH_2_CHOHCH_3_; **SBE-β****-CD:** R = -(CH_2_)_4_SO_3_-Na^+^.

**Figure 3 molecules-27-05600-f003:**
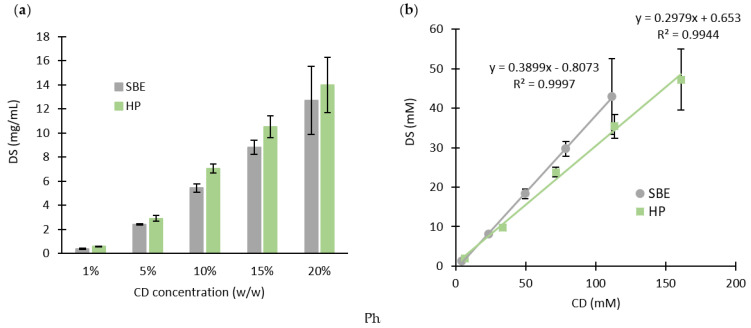
(**a**) DS solubility in assorted CD solutions; (**b**) phase solubility diagrams of DS and CD (mean ± SD, *n* = 3).

**Figure 4 molecules-27-05600-f004:**
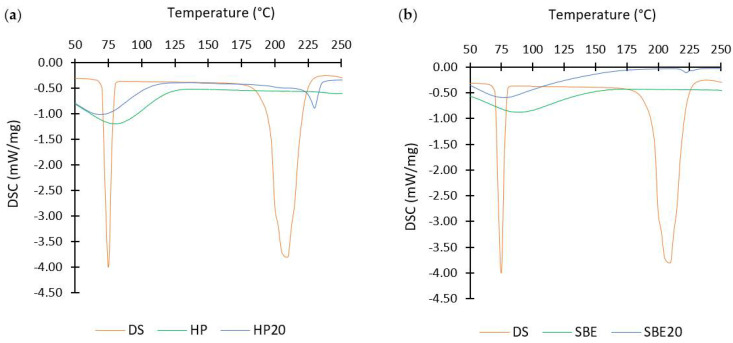
Differential Scanning Calorimetry (DSC) thermographs of (**a**) DS, HP-CD, and freeze-dried formulation (HP20); (**b**) DS, SBE-CD, and the freeze-dried formulation SBE20.

**Figure 5 molecules-27-05600-f005:**
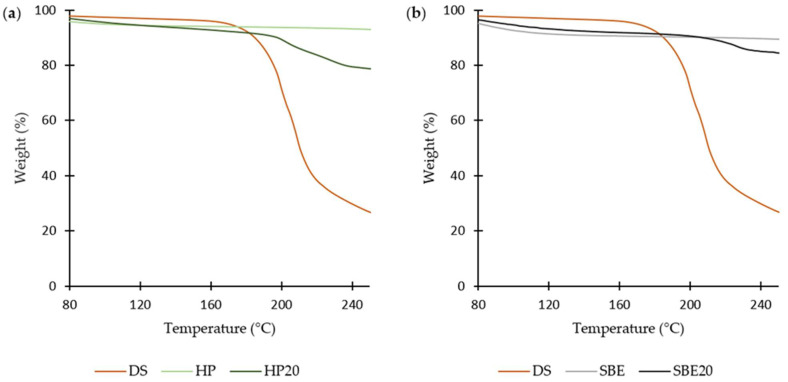
Thermogravimetric Analysis (TGA) thermographs of (**a**) DS, HP-CD, and HP-CD freeze-dried formulation; (**b**) DS, SBE-CD, and SBE-CD freeze-dried formulation.

**Figure 6 molecules-27-05600-f006:**
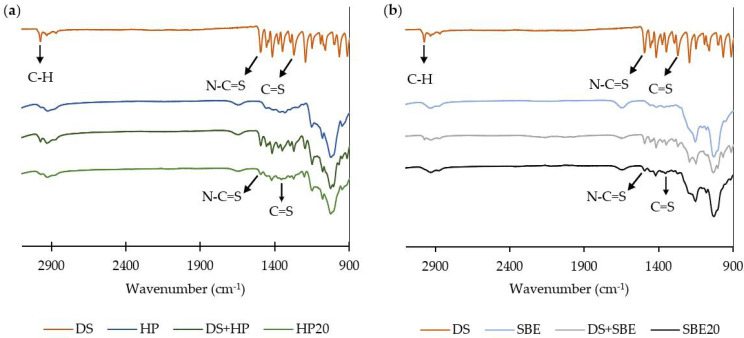
Fourier Transform Infrared Spectroscopy (FTIR) spectra of (**a**) DS, physical mixture of DS with HP-CD, raw HP-CD, and HP-CD freeze-dried formulation; (**b**) DS, physical mixture of DS with SBE-CD, raw SBE-CD, and SBE-CD freeze-dried formulation.

**Figure 7 molecules-27-05600-f007:**
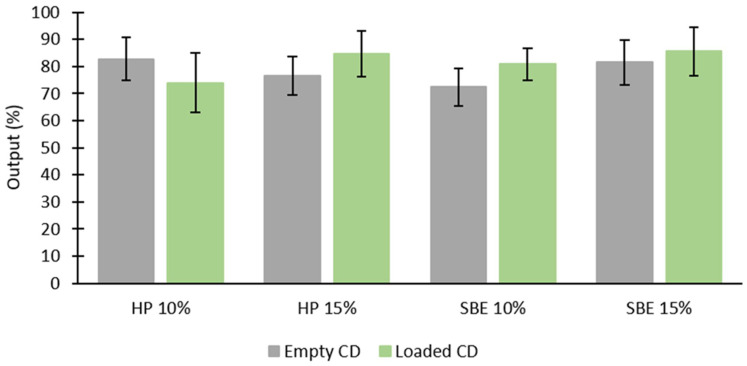
Aerosol output of each empty CD (10% and 15%), and each CD (10% and 15%) loaded with DS (mean ± SD, *n* = 3).

**Figure 8 molecules-27-05600-f008:**
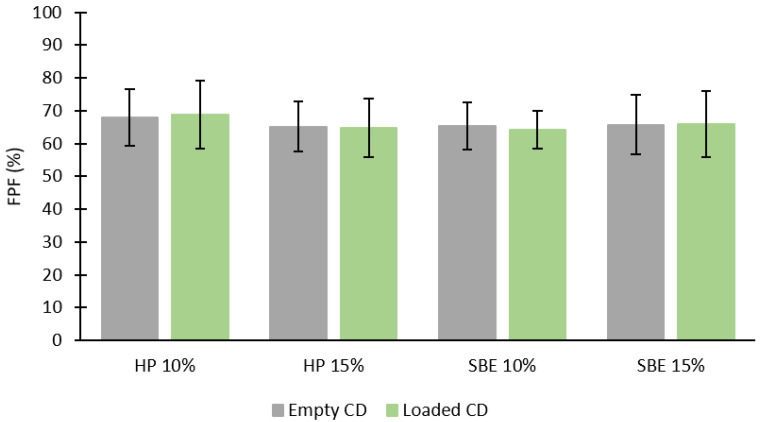
FPF of each empty CD (10% and 15%), and each CD (10% and 15%) loaded with DS (mean ± SD, *n* = 3).

**Table 1 molecules-27-05600-t001:** Stability constants (K_1:1_), using both S_0_ and S_int_, and complexation efficiency (CE) of DS with each CD.

CD	K_1:1_ (M^−1^)		CE
	Using S_0_	Using S_int_	
SBE	149.3	−1263 ^a^	0.64
HP	224.9	1539	0.42

^a^ S_int_ < 0.

**Table 2 molecules-27-05600-t002:** VMD, Span, and % ≤ 5.4 µm of each empty CD (10% and 15%), and each CD (10% and 15%) loaded with DS (mean ± SD, *n* = 3).

Parameter	Formulation	Empty Cyclodextrin	Loaded Cyclodextrin
VMD (µm)	HP 10%	1.8 ± 1	2.52 ± 0.61
	HP 15%	1.49 ± 0.69	3.37 ± 0.81 ^a^
	SBE 10%	1.67 ± 1.23	3 ± 0.99
	SBE 15%	1.63 ± 1	2.95 ± 0.84
Span	HP 10%	4.96 ± 3.61	1.41 ± 0.24
	HP 15%	5.28 ± 2.85	2.12 ± 0.24
	SBE 10%	3.57 ± 0.48	2.3 ± 0.38
	SBE 15%	6.51 ± 4.2	2.47 ± 0.31
% ≤ 5.4 µm	HP 10%	82.16 ± 2.15	92.95 ± 8.03
	HP 15%	84.14 ± 2.16	76.28 ± 10.86
	SBE 10%	90.20 ± 10	79.42 ± 8.02
	SBE 15%	80.53 ± 6.35	77.04 ± 10.61

^a^*p* < 0.05.

## Data Availability

The data presented in this study are available in this article. Enquiries may be made to the corresponding author.

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
