# Peer review of "Preparation and Characterization of Disulfiram and Beta Cyclodextrin Inclusion Complexes for Potential Application in the Treatment of SARS-CoV-2 via Nebulization"

_molecules, 2022, doi:10.3390/molecules27175600_

Round 1

Reviewer 1 Report

The manuscript by Najlah and coworkers reports on the supramolecular host-guest complexation between beta-cyclodextrin (CDs) derivatives and disulfiram (DS). The solubility of disulfiram in water is enhanced upon the complexation with the CDs. In addition, the CD-DS solutions were also characterized for their application as inhalable aerosols using the nebulization method. Overall, this work has a certain novelty and necessary evidence and discussion. I think the manuscript is suitable for publication after some improvements.

11) The molar absorptivity of disulfiram was used to determine its water solubility, however, it is known that complexation affect the molar absorptivity of the encapsulated guest. Therefore, the authors should make sure that the molar absorptivity does not change significantly, otherwise the non-linear response observed in Figure 3b with HP-CD might be due to the different molar absorptivity of the free DS compared to the complexed one.

22)    Although, most experiments support the formation of supramolecular complexes, they do not provide a clear evidence for the formation of inclusion-type of complexation; this should be discussed in the manuscript.

33)    To further guide the reader, the characteristic vibration bands in Figure 6 should be labeled.

44)    Line 210: change “spectrums” to “spectra”.

Author Response

Authors: we thank the reviewer for their constructive feedback

11) The molar absorptivity of disulfiram was used to determine its water solubility, however, it is known that complexation affect the molar absorptivity of the encapsulated guest. Therefore, the authors should make sure that the molar absorptivity does not change significantly, otherwise the non-linear response observed in Figure 3b with HP-CD might be due to the different molar absorptivity of the free DS compared to the complexed one.

We thank you reviewer for the valid point. Actually, during the method development stage of this work, we had similar concerns about the molar absorptivity of the drug after complexation. Therefore, we have run UV scanning of the drug at different concentrations of CD to prove that there was no change in lambda max. In addition, we confirmed the accuracy of the calibration curves of DS in increased concentrations of either of both cyclodextrins using HPLC. We addressed this in the method section on our methods section.

22)    Although, most experiments support the formation of supramolecular complexes, they do not provide a clear evidence for the formation of inclusion-type of complexation; this should be discussed in the manuscript.

We thank the reviewer for their comment. We have clarified this in our manuscript.

33)    To further guide the reader, the characteristic vibration bands in Figure 6 should be labelled.

The characteristic vibration bands are now added to Figure 6

44)    Line 210: change “spectrums” to “spectra”.

We thank the reviewer for spotting this error which we have rectified as required.

Reviewer 2 Report

The article titled  Preparation and Characterization of Disulfiram and Beta Cy- 2

clodextrin Inclusion Complexes for the Treatment of SARS-CoV-2 Via Nebulization after consideration of minor comments.

.

1)      Abstract, authors should mention change in activity against Cov-2

2)      Introduction, authors displayed how they can improve the solubility but what about the second problem rapid metabolism instability.

3)      The authors depended mainly on FTIR but also NMR is important for structure elucidation .

4)      Also, the effect on biological activity after complex formation should be assessed.

Author Response

We thank the reviewer for their constructive comments

  • Abstract, authors should mention change in activity against Cov-2

We thank the reviewer for their comment. Unfortunately, we don’t have the facilities to test our formulations against Cov-2 in vitro. Therefore, we are not able to add such information to the abstract.  

  • Introduction, authors displayed how they can improve the solubility but what about the second problem rapid metabolism instability.

We thank the reviewer for the valid comment.  We have proposed the pulmonary route of administration (inhalation) to provide a solution for rapid metabolism.

  • The authors depended mainly on FTIR but also NMR is important for structure elucidation.

We thank the reviewer for their comment. We have highlighted the need for NMR for further characterisation in our manuscript. At this stage, we are unable to perform such characterisation.

4)      Also, the effect on biological activity after complex formation should be assessed.

As described above, the manuscript describes a method to enhance disulfiram solubility for a proposed anti-viral application. We have added the suggestion of exploring biological activity after complexation as future work.

Reviewer 3 Report

this is a low quality study of complexation of disulfiram with cyclodextrins in which it is not acceptable to indicate its potential use in the treatment of 

Author Response

We fully respect the reviewer's opinion, we believe that the manuscript presents a novel idea of using disulfiram, solubilised by cyclodextrin, to proposedly treat via the inhalation route of administration.

Round 2

Reviewer 3 Report

some analytical techniques, such as X Ray diffraction, must be employed to the confirmación of disulfiram inclusion into CD cavities. Inaddition, the constant solubitity values mightbe calculated.              This research isnot related to a SARS COV 2 treatment as is mentioned  by authors

Author Response

We have responded to all comments in the attached cover letter
